# Lazertinib versus Platinum-Based Chemotherapy with Epidermal Growth Factor Receptor (EGFR)-Positive Non-Small-Cell Lung Cancer after Failing EGFR-Tyrosine Kinase Inhibitor: A Real-World External Comparator Study

**DOI:** 10.3390/cancers16122169

**Published:** 2024-06-07

**Authors:** Junho Lee, Hyesung Lee, Dongwon Yoon, Eun-Young Choi, Jieun Woo, Bobae Jo, Sohee Kim, Ju-Young Shin, Hyun Ae Jung

**Affiliations:** 1Division of Hematology and Oncology, Department of Medicine, Samsung Medical Center, Sungkyunkwan University School of Medicine, Seoul 06351, Republic of Korea; klao0403@naver.com; 2School of Pharmacy, Sungkyunkwan University, Suwon 16419, Republic of Korea; gul2@skku.edu (H.L.); dwyoon09@gmail.com (D.Y.); eychoi301@g.skku.edu (E.-Y.C.); 3Department of Biohealth Regulatory Science, Sungkyunkwan University, Suwon 16419, Republic of Korea; jejjy04222@g.skku.edu; 4Yuhan Corporation, Seoul 06927, Republic of Korea; bbjo@yuhan.co.kr (B.J.); sohee.kim@yuhan.co.kr (S.K.); 5Department of Clinical Research Design & Evaluation, Samsung Advanced Institute for Health Sciences and Technology (SAIHST), Sungkyunkwan University, Seoul 06351, Republic of Korea

**Keywords:** external comparator study, NSCLC, EGFR, lazertinib, platinum-based chemotherapy

## Abstract

**Simple Summary:**

Lazertinib, a third-generation EGFR-TKI, selectively inhibits both the common EGFR mutation and the T790M mutation in NSCLC patients. No previous studies have compared lazertinib with platinum-based chemotherapy. In this external control retrospective study, we have compared the efficacy of lazertinib and platinum-based chemotherapy in patients who had previously received EGFR-TKI treatment. This study included 200 patients from the LASER 201, LASER 301, and LASER-PMS studies, and 334 patients who received platinum-based chemotherapy after prior EGFR-TKI treatment for SMC. After propensity score matching, we selected 156 patients from each group. The PFS was significantly longer in those patients treated with lazertinib than in those treated with platinum-based chemotherapy (10.97 months vs. 5.10 months) after PSM. Lazertinib also demonstrated superior OS, ORR, and TTD compared to platinum-based chemotherapy. Based on this retrospective, external control study, lazertinib demonstrated significantly superior efficacy compared to platinum-based chemotherapy. The external controls provide important context to evaluate efficacy in single-arm studies.

**Abstract:**

Background: Lazertinib is a third-generation tyrosine kinase inhibitor of epidermal growth factor receptor (EGFR-TKI) that selectively inhibit common EGFR mutation and T790M mutation in non-small-cell lung cancer (NSCLC) patients. No previous studies have compared lazertinib to platinum-based chemotherapy. We have compared lazertinib with platinum-based chemotherapy in EGFR-mutated NSCLC patients after previous EGFR-TKI therapy. Methods: We retrospectively compared 200 patients from LASER201, LASER301, and LASER-PMS studies to 334 patients who were treated with platinum-based chemotherapy after previous EGFR-TKI from the Samsung Medical Center. After propensity score matching (PSM), we selected 156 patients from each group. The primary outcome was progression-free survival (PFS), with overall survival (OS), objective response rate (ORR), and time to treatment discontinuation (TTD) as secondary outcomes. Results: The median follow-up of PFS was 15.61 months in the lazertinib group and 21.67 months in the external control group. The PFS was significantly longer in patients who were treated with lazertinib than those treated with platinum-based chemotherapy (10.97 months vs. 5.10 months; adjusted hazard ratio (HR) 0.40; 95% confidence interval (CI), 0.29–0.55; *p* < 0.01) after PSM. Lazertinib showed superior OS (32.23 months vs. 18.73 months; adjusted HR 0.45; 95% CI, 0.29–0.69; *p* < 0.001), ORR (64.1% vs. 47.4%), and TTD (11.66 months vs. 6.73 months; adjusted HR 0.54; 95% CI, 0.39–0.75; *p* < 0.001) compared to platinum-based chemotherapy. Conclusion: Based on this retrospective, external control study, lazertinib has demonstrated significantly better efficacy compared with platinum-based chemotherapy. The external controls provide important context to evaluate efficacy in single-arm studies.

## 1. Introduction

Lazertinib, an oral third-generation tyrosine inhibitor (TKI) of the epidermal growth factor receptor kinase (EGFR), has demonstrated its efficacy in non-small-cell lung cancer (NSCLC) patients with exon 19 deletion, L858R mutation, and T790M mutation [1,2]. The LASER 201 study (NCT03046992) is a single-arm, phase I/II study that enrolled advanced NSCLC patients who were pretreated with EGFR-TKI and reported median progression-free survival (PFS) of 11.1 and for whom overall survival (OS) was not reached [1]. Based on a recent update from LASER301 (NCT04248829), which is a double-blind, randomized, international phase III study comparing lazertinib to gefitinib as the first-line treatment for advanced NSCLC patients with EGFR mutation, the lazertinib group demonstrated that the median PFS was 20.6 months. The median PFS in the lazertinib group was more than double that of the median PFS of 9.7 months in the gefitinib group (hazard ratio (HR): 0.45) [2]. According to the AURA3 study, in comparison with platinum-based chemotherapy, osimertinib is the gold standard of cancer treatment in patients with T790M-positive NSCLC who experienced disease progression on first- or second-generation EGFR-TKI [3].

To the best of our knowledge, no randomized controlled trials comparing lazertinib to platinum-based chemotherapy or osimertinib in patients who have failed first- or second-generation EGFR-TKIs have been performed. Ideally, a study comparing lazertinib to platinum-based chemotherapy, akin to the AURA3 study, or a study of a direct comparison of lazertinib to osimertinib in patients who are harboring a T790M-positive mutation is needed to establish the superiority of lazertinib over platinum-based chemotherapy or comparable efficacy to osimertinib. However, given that osimertinib is currently the standard treatment for patients with T790M mutations, such comparative studies are ethically and economically challenging.

In this context, the difficulties of comparing lazertinib with platinum-based chemotherapy may be addressed by establishing an external comparator in real-world settings. An external control is a comparator arm with no treatment or standard of care drawn from another population in different settings. Since the background rates of outcomes cannot be adjusted in uncontrolled single-arm studies, a single-arm study with external control is considered to have a higher quality of evidence than those without [4,5].

We conducted an external control study to compare the effectiveness of lazertinib to that of platinum-based chemotherapy using individual patient data from the LASER201 study, LASER 301 study, and LASER-PMS study, and the patient registry of a tertiary hospital in South Korea.

## 2. Patients and Methods

### 2.1. Data Sources

We retrieved individual-patient-level data for the lazertinib group from the LASER 201, LASER 301, and LASER PMS studies, which included advanced NSCLC patients who were harboring the EGFR mutation. Among the patients included in the LASER 301 study, we identified those who had switched from gefitinib to lazertinib after confirming disease progression and a T790M-mutation-positive status (crossover group) as the study population of interest. The LASER-PMS study is a post-marketing surveillance study based on the Pharmaceutical Affairs Act in South Korea. The real-world individual-patient-level data for the external comparator group were retrieved from the Samsung Medical Center-Clinical Data Warehouse (SMC-CDW). This database, named ROOT-HEALTH, includes all information on the demographics, clinical characteristics, biomarkers, treatments, and clinical outcomes for patients who have visited the Samsung Medical Center [6]. We collected variables from the SMC-CDW based on the data collected from the LASER 201, LASER 301, and LASER PMS studies.

### 2.2. Study Population and Exposure

Among patients ≥ 18 years of age diagnosed with advanced NSCLC between 1 June 2015 and 30 April 2022, the lazertinib group comprised patients diagnosed with EGFR T790M-mutation-positive NSCLC after failing first- or second-generation EGFR-TKI; who participated in any of the LASER 201, LASER 301, or LASER-PMS studies; and who were treated with lazertinib between 27 November 2017 and 30 April 2022. The external control group included NSCLC patients who harbored the EGFR mutation and were treated with platinum-based chemotherapy at the Samsung Medical Center after failing treatment of first- or second-generation EGFR-TKI during the same time frame. The index date was defined as the first date of lazertinib treatment for the lazertinib group and the first date of platinum-based chemotherapy treatment for the external comparator group. All eligible patients scored 0 or 1 in Eastern Cooperative Oncology Group (ECOG) performance status. We excluded patients who needed to be treated for or had a history of treatment for brain metastases within 14 days preceding the index date and patients who needed to be treated for or had a history of treatment for double primary cancer before the index date.

### 2.3. Outcomes

The primary outcome was PFS, as determined by investigator assessment, and the patients were followed from the index date to the date of progression, the date of all-cause death within 14 weeks from the last date of assessment, or the end date of the study period (20 March 2023), whichever came first. The secondary outcomes were OS, objective response rate (ORR), disease control rate (DCR), and time to treatment discontinuation (TTD). The OS, ORR, and DCR were determined by investigator assessment, and OS was defined as the time interval from the index date to the date of all-cause death or the date of the last follow-up visit. The ORR consisted of complete or partial response, and the DCR consisted of complete response, partial response, or stable disease. The TTD of the lazertinib group was defined as the time interval from the index date to the last date of lazertinib treatment, and that of the external comparator group was defined as the time interval from the index date to the date 21 days from the last platinum-based chemotherapy treatment.

### 2.4. Potential Confounders

Age and sex were included as demographic characteristics, and clinical characteristics included smoking status, ECOG performance status, tumor histology, American Joint Committee on Cancer Tumor-Node-Metastases stage, and brain metastasis. The biomarker-related characteristics included EGFR mutation status, and the cancer-treatment-related characteristics included previous lines of systemic therapy, previous lines of EGFR-TKI, type of previous EGFR-TKI, time from immediate previous EGFR-TKI, and duration of previous EGFR-TKI. The demographic characteristics were assessed at the index date, and other characteristics were assessed before the index date.

### 2.5. Statistical Analysis

We summarized the baseline characteristics using medians with Q1 and Q3 (or min and max) and counts with proportions for continuous and categorical variables. To balance the lazertinib group and the external comparator group in baseline characteristics, we used 1:1 propensity score matching (PSM), which is based on the greedy nearest neighbor matching algorithm without replacement [7]. Using a logistic regression model from age, sex, smoking status, ECOG performance status, brain metastasis, previous lines of EGFR-TKI treatment, time from immediate previous EGFR-TKI treatment, and the duration of previous EGFR-TKI treatment, the propensity scores were calculated. The absolute standardized difference (aSD) was used to determine significant imbalances between exposure groups.

For the time-to-event outcomes (PFS, OS, and TTD), we first calculated the incidence rate per 100 person-years with a 95% confidence interval (CI). The Kaplan–Meier (KM) method was used to estimate the median survival time with a 95% CI, and the log-rank test was used to compare the KM curve between the lazertinib and the external comparator groups. Crude and adjusted HRs with 95% CIs were estimated using a Cox proportional hazard model. For the ORR and DCR, we calculated the percentages with 95% CIs, and the crude and adjusted odds ratios (OR) with 95% CIs were estimated using a logistic regression model. We conducted subgroup analyses by age group (<65 or ≥65 years), sex, and smoking status to investigate differences in the comparative effectiveness of the primary and secondary outcomes by the subgroups mentioned above.

We performed several sensitivity analyses for primary and secondary outcomes to assess the validity of our main findings. First, we applied inverse probability of treatment weighting (IPTW) to address imbalances of potential confounders between the lazertinib and the external comparator groups. Second, we restricted the lazertinib groups to patients who participated in the LASER 201 and LASER 301 studies, as the data characteristics between the clinical trials and the PMS studies may differ and induce potential bias. Third, we additionally constructed another two lists of covariates for estimating PS to investigate the potential impact of different modeling on our study (Appendix A).

All analyses were performed using SAS 9.4 (SAS Institute Inc., Cary, NC, USA), and statistical significance was defined at a level of 0.05. Informed consent was waived, as the study utilized anonymized data. This study was approved by the Institutional Review Board of the Samsung Medical Center, South Korea (No. 2023-04-096), and conducted in accordance with the Declaration of Helsinki.

## 3. Results

### 3.1. Study Population and Clinical Characteristics

Figure 1 describes the selection of the study population. Between 27 November 2017 and 30 April 2022, 200 patients were enrolled in either the LASER 201, LASER 301, or LASER-PMS studies and received lazertinib treatment. During the same period, 334 other patients received platinum-based chemotherapy at the Samsung Medical Center. The median patient age was 64 years in the lazertinib group and 63 years in the external control group, and 57% of the patients in the lazertinib group and 57.2% of the patients in the external control group were female. All of the patients in the lazertinib group had T790M mutations and had failed treatment of first- or second-generation EGFR-TKIs. In the external control group, 43.4% of patients had the EGFR L858R mutation, 53.6% of patients had EGFR exon 19 deletion, and 5.7% of patients had other EGFR mutations, including L861Q and G719X. After PS matching, 156 patients from both groups were matched 1:1, and the distribution of covariates was similar in the two groups (Table 1).

### 3.2. Progression-Free Survival and Overall Survival

The median follow-up duration of PFS was 15.61 months (inter-quartile range (IQR); 10.81–49.54 months) in the lazertinib group and 21.67 months (IQR; 18.17–34.13 months) in the external control group. The median PFS was 12.16 months (95% CI: 9.72–13.63 months) in the lazertinib group and 5.03 months (95% CI: 4.80–5.60 months) in the external control group (adjusted HR = 0.40, 95% CI: 0.31–0.53, *p* < 0.001). The median OS was 33.84 months (95% CI: 30.13 months not reached) in the lazertinib group and 20.00 months (95% CI: 17.03–23.90 months) in the external control group (adjusted HR = 0.48, 95% CI: 0.33–0.70, *p* < 0.001).

After PS matching, the median PFS was 10.97 months (95% CI: 8.77–12.52) in the lazertinib group and 5.10 months (95% CI: 4.67–6.07 months) in the external control group (adjusted HR = 0.40, 95% CI: 0.29–0.55, *p* < 0.001). The 6-month PFS rate in the lazertinib group was 74.8% (95% CI: 67.1–81.0%), and the 12-month PFS rate was 47.4% (95% CI: 38.9–55.5%). The 6-month PFS rate in the external control group was 43.2% (95% CI: 35.3–50.9%), and the 12-month PFS rate was 13.3% (95% CI: 8.4–19.3%) (Figure 2A, Appendix A). The median OS was 32.23 months (95% CI: 29.08 months not reached) in the lazertinib group and 18.73 months (95% CI: 14.40–24.87 months) in the external control group (adjusted HR = 0.45, 95% CI: 0.29–0.69, *p* < 0.001). The 12-month OS rate in the lazertinib group was 84.4% (95% CI: 76.7–89.7%), and the 36-month OS rate was 42.2% (95% CI: 26.5–57.0%). The 12-month OS rate in the external control group was 68.3% (95% CI: 60.0–74.9%), and the 36-month OS rate was 27.0% (95% CI: 18.0–36.8%) (Figure 2B, Appendix A). Similar findings with the overall population were observed in the subgroup and sensitivity analyses (Table 2).

### 3.3. Objective Response Rate

The ORR was 65.5% (95% CI: 58.9–72.1%) in the lazertinib group and 47.0% (95% CI: 41.7–52.4%) in the external control group (adjusted OR = 1.89, 95% CI: 1.18–3.04, *p* = 0.009). The disease control rate (DCR) was 88.0% (95% CI: 83.5–92.5%) in the lazertinib group and 75.8% (95% CI: 71.2–80.4%) in the external control group. After applying PSM, the ORR was 64.1% (95% CI: 56.6–71.6%) in the lazertinib group and 47.4% (95% CI: 39.6–55.3%) in the external control group (adjusted OR = 1.92 (95% CI: 1.08–3.39, *p* = 0.026)). The DCR was 88.5% (95% CI: 83.5–93.5%) in the lazertinib group and 77.6% (95% CI: 71.0–84.1%) in the external control group (Table 3).

### 3.4. Time to Treatment Discontinuation

At the date of data cut-off, 151 patients (75.5%) in the lazertinib group had discontinued lazertinib treatment and 266 patients (79.6%) in the external control group had discontinued platinum-based chemotherapy. The median TTD was 12.32 months (95% CI: 10.41–13.83) in the lazertinib group and 6.53 months (95% CI: 5.73–6.97) in the external control group. After applying PSM, TTD was 11.66 months (95% CI: 9.10–13.83) in the lazertinib group and 6.73 months (95% CI: 5.50–7.10) in the external control group. The lazertinib group demonstrated longer TTD than the external control group both before and after applying PSM according to the Kaplan–Meier curve of time to treatment discontinuation (Figure 2C and Appendix A).

## 4. Discussion

As far as we know, this is the first study to show that lazertinib was associated with superior PFS, OS, ORR, and TTD, compared with platinum-based chemotherapy in patients who progressed on first- or second-generation EGFR TKIs. The median PFS of the lazertinib group was 10.97 months, and numerically similar to PFS for osimertinib in the AURA3 study, which was 10.1 months. The hazard ratio for PFS was 0.40 (*p* < 0.001), indicating the significantly superior efficacy of lazertinib compared to platinum-based chemotherapy. These results suggest that lazertinib is comparable with osimertinib as a therapeutic option for patients with EGFR-T790M-mutated NSCLC [3]. The lazertinib group showed significantly longer median OS (32.23 months versus 18.73 months) compared to the control group, with a hazard ratio of 0.45 (*p* < 0.001). Unlike this result, osimertinib did not show a significant benefit for OS in the AURA3 study, which might be due to the high rate of treatment switch to osimertinib after disease progression in the control group [8].

Lazertinib is a third-generation TKI designed to be highly selective for the EGFR-T790M mutation. According to the LASER 201, LASER 301, and LASER-PMS studies, lazertinib has higher efficacy for NSCLC patients with both the EGFR mutation and the T790M mutation treated with previous first- or second-generation EGFR-TKIs [1,2]. However, comparisons with lazertinib and platinum-based chemotherapy for NSCLC patients with the EGFR mutation after progression during the treatment of early generation EGFR TKIs had not previously been performed. Based on the AURA3 study, the standard treatment of EGFR-T790M-mutated NSCLC patients after failing first- or second-generation EGFR-TKIs is osimertinib. Therefore, randomized controlled trials comparing lazertinib with platinum-based chemotherapy are difficult to conduct due to ethical and economic conflicts [9].

To address these problems, we conducted the present study using an external comparator group. In addition, the external control designs also contribute to more cost-effective, faster, and more ethical clinical trials. Since the number of accelerated approvals based on the results of single-arm studies have increased, studies using external real-world control groups have been conducted to complement those results [10].

The present study was conducted to assess the effectiveness of lazertinib and to provide supportive evidence to the Ministry of Food and Drug Safety in South Korea (MFDS). The regulatory standards require substantial evidence of effectiveness derived from adequate and well-controlled trials that typically utilize a valid comparison to an internal concurrent control. However, when it is impractical or unethical for an internal control, external controls may be an acceptable alternative [11]. Real-world studies can yield comparable conclusions to randomized controlled trials when the study design and measurements closely emulate controlled trials, accounting for differences, chance, and residual confounding factors [12]. The key consideration when adopting an external comparator design is the exchangeability of patients in the real-world setting to those in trials, which could be a source of bias if not met. The following six factors, mentioned by U.S. FDA regulators, can be used to evaluate exchangeability: (1) eligibility criteria, (2) distribution of baseline characteristics, (3) treatment comparability, (4) method used for evaluating outcomes, (5) time frame for data collection, and (6) data collection in the same setting by the same investigators [13]. By obtaining complete individual-patient-level data, we could emulate the eligibility criteria in the external comparator group corresponding to the lazertinib group as closely as possible, resulting in minimal selection bias. The PSM allows researchers to minimize confounding bias by balancing the distribution of potential confounders [14]. In addition, the same time frame for patient selection and the same variable measurement were applied to both groups to minimize surveillance bias and unmeasured confounding bias.

This study has several limitations. First, most patients in the external group were EGFR-T790M-negative or unknown, unlike the patients of the lazertinib group. When we designed this external control study, we considered T790M-mutation-positive NSCLC patients after failure of a first- or second-generation EGFR TKI as the external control group. Because the standard treatment for T790M-mutated NSCLC patients has been osimertinib during the period of clinical studies for lazertinib, few patients who were T790M-mutation-positive were treated with platinum-based chemotherapy. The number was too small to conduct an external study. However, in previous retrospective studies of second-line therapy for EGFR-mutated NSCLC and IMPRESS, in which most patients were EGFR-T790M-negative, the median PFS for platinum-based chemotherapy was about 3–5.5 months [15,16,17]. In the AURA3 study, which included EGFR-T790M-positive patients, the median PFS for platinum-based chemotherapy after first-line gefitinib in EGFR-mutated NSCLC was 4.4 months [3]. The above survival outcomes have shown similar PFS for platinum-based chemotherapy after failure of a prior EGFR TKI, regardless of T790M mutation status. It was also similar to the median PFS for the external control group from this study (5.03 months; 95% CI: 4.80–5.60 months). In addition, a retrospective study comparing platinum-based chemotherapy between T790M-positive and T790M-negative NSCLC patients after EGFR-TKI failure showed no differences in clinical outcome between the two groups [18]. We collected data as an external control group regardless of the EGFR T790M mutation status on the basis that EGFR T790M status does not affect the efficacy of platinum-based chemotherapy after failure of first- or second-generation EGFR-TKI treatment.

Second, the distribution of previously used afatinib and gefitinib between the lazertinib group (gefitinib vs. afatinib, 60.3% vs. 29.5%) and the external comparator (gefitinib vs. afatinib, 34.6% vs. 57.1%) in this study is quite different. According to the LUX-LUNG 7 trial, which compared the efficacy of afatinib and gefitinib as a first-line treatment in EGFR-mutated NSCLC patients, there was no OS difference (median 27.9 months vs. 24.5 months, HR 0.86, 95% CI: 0.66–1.12). The PFS was also similar between afatinib and gefitinib (median 11.0 months and 10.9 months) [19]. Considering the OS and PFS from the LUX-LUNG 7 trial, we suggest that the differences in the distribution of prior EGFR-TKI had a minimal impact on the efficacy of the subsequent treatment.

Unmeasured confounding bias due to differences in data collection settings is the greatest challenge when using external comparators, and it is not easily addressed in conventional ways. New methodological or statistical approaches should be proposed to overcome this issue.

## 5. Conclusions

Lazertinib showed superior PFS, overall survival, ORR, and TTD compared to platinum-based chemotherapy in an external comparator group before and after PSM.

## Figures and Tables

**Figure 1 cancers-16-02169-f001:**
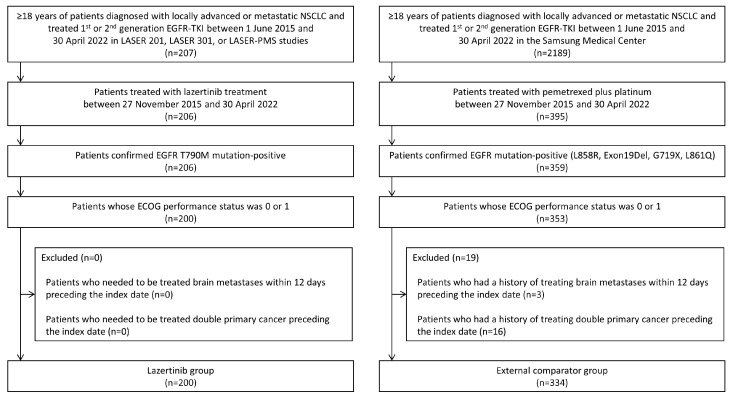
Selection of the study population.

**Figure 2 cancers-16-02169-f002:**
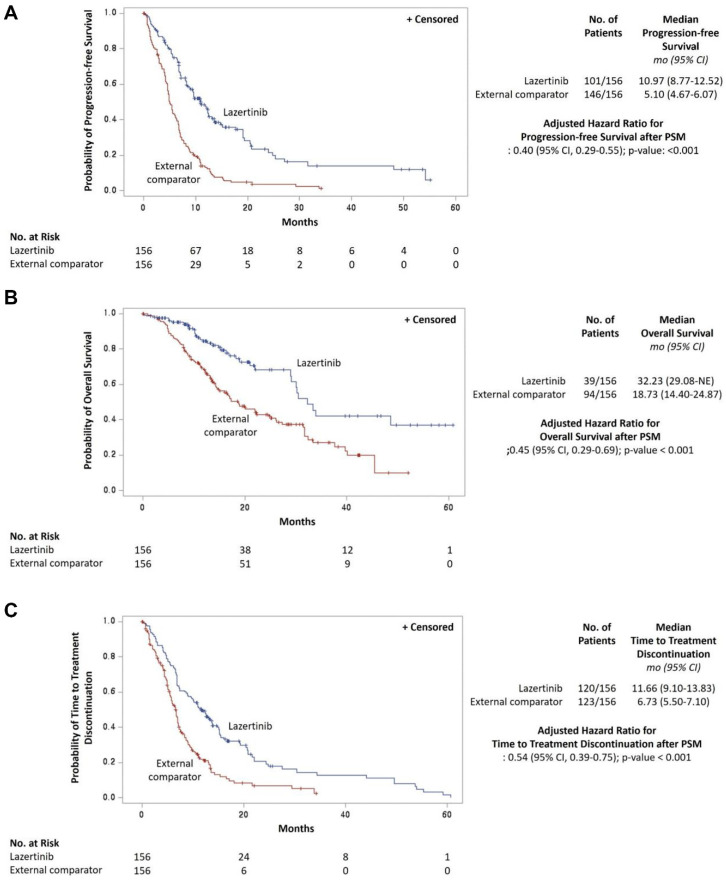
(**A**) Progression-free survival after propensity score matching. (**B**) Overall survival after propensity score matching. (**C**) Time to treatment discontinuation after propensity score matching. PS, propensity score.

**Table 1 cancers-16-02169-t001:** Baseline characteristics.

	Before PS Matching			After PS Matching		
	Lazertinib (*n* = 200)	External Comparator (*n* = 334)	aSD	Lazertinib (*n* = 156)	External Comparator (*n* = 156)	aSD
Age, median (Q1–Q3)	64 (56–71)	63 (56–70)	0.11	63.5 (55–70.5)	62.5 (57–69)	0.03
Female sex, *n* (%)	114 (57.0)	191 (57.2)	0.00	85 (54.5)	87 (55.8)	0.03
No history of smoking, *n* (%)	123 (61.5)	213 (63.8)	0.05	94 (60.3)	94 (60.3)	0.00
ECOG performance status, *n* (%)						
0	41 (20.5)	19 (5.7)	0.45	16 (10.3)	17 (10.9)	0.02
1	159 (79.5)	315 (94.3)		140 (89.7)	139 (89.1)	
Adenocarcinoma tumor histology, *n* (%)	194 (97.0)	320 (95.8)	0.06	151 (96.8)	150 (96.1)	0.03
Brain metastasis, *n* (%)	76 (38.0)	90 (27.0)	0.24	56 (35.9)	56 (35.9)	0.00
EGFR mutation status, *n* (%)						
L858R	37 (18.5)	145 (43.4)	0.56	27 (17.3)	67 (43.0)	0.58
Exon19Del	80 (40.0)	179 (53.6)	0.27	68 (43.6)	83 (53.2)	0.19
Others (G719X, L861Q, others, unknown)	83 (41.5)	19 (5.7)	0.35	61 (39.1)	8 (5.1)	0.90
Previous lines of systemic therapy, median (min–max)	1 (1–10)	1 (1–5)	0.45	1 (1–5)	1 (1–5)	0.06
Previous lines of EGFR-TKI treatment, median (min–max)	1 (1–10)	1 (1–5)	0.36	1 (1–3)	1 (1–5)	0.03
Type of previous EGFR-TKI treatment, median (min–max)						
Gefitinib	120 (60.0)	117 (35.0)	0.52	94 (60.3)	54 (34.6)	0.53
Erlotinib	25 (12.5)	77 (23.1)	0.28	15 (9.6)	43 (27.6)	0.47
Afatinib	61 (30.5)	193 (57.8)	0.57	46 (29.5)	89 (57.1)	0.58
Time from immediate previous EGFR-TKI treatment, *n* (%)			0.58			0.15
<30 days	140 (70.0)	167 (50.0)		108 (69.2)	117 (75.0)	
≥30 days	47 (23.5)	165 (49.4)		45 (28.9)	37 (23.7)	
No	13 (6.5)	2 (0.6)		3 (1.9)	2 (1.3)	
Duration of previous EGFR-TKI treatment, *n* (%)			0.45			0.14
<6 months	8 (4.0)	59 (17.7)		6 (3.9)	11 (7.1)	
≥6 months	192 (96.0)	275 (82.3)		150 (96.2)	145 (93.0)	

PS, propensity score; SD, standardized difference; ECOG, Eastern Cooperative Oncology Group; EGFR, epidermal growth factor receptor; EGFR-TKI, epidermal growth factor receptor-tyrosine kinase inhibitor.

**Table 2 cancers-16-02169-t002:** Results of subgroup and sensitivity analyses after propensity score matching.

	Adjusted HR (95% CI) ^a^			Adjusted OR (95% CI) ^a^	
	PFS	OS	TTD	ORR	DCR
**Subgroup analyses**					
Overall	0.40 (0.29–0.55)	0.45 (0.29–0.69)	0.54 (0.39–0.75)	1.92 (1.08–3.39)	1.96 (0.93–4.13)
Age group					
<65 years	0.40 (0.25–0.62)	0.50 (0.27–0.91)	0.55 (0.35–0.88)	1.45 (0.66–3.17)	0.98 (0.36–2.71)
≥65 years	0.40 (0.23–0.68)	0.68 (0.34–1.36)	0.74 (0.44–1.24)	2.00 (0.81–4.98)	2.16 (0.68–6.88)
Sex					
Male	0.32 (0.18–0.58)	0.36 (0.16–0.83)	0.52 (0.29–0.96)	3.05 (1.16–8.03)	4.04 (0.91–18.00)
Female	0.44 (0.28–0.69)	0.65 (0.36–1.17)	0.58 (0.37–0.92)	1.81 (0.80–4.08)	2.53 (0.92–6.94)
Smoking status					
Ever smoker	0.38 (0.20–0.72)	0.36 (0.15–0.84)	0.54 (0.28–1.04)	2.60 (0.87–7.75)	1.94 (0.46–8.20)
Never smoker	0.34 (0.22–0.51)	0.55 (0.32–0.93)	0.48 (0.32–0.73)	2.06 (0.99–4.29)	2.70 (1.02–7.11)
**Sensitivity analyses**					
Main analyses ^b^	0.40 (0.29–0.55)	0.45 (0.29–0.69)	0.54 (0.39–0.75)	1.92 (1.08–3.39)	1.96 (0.93–4.13)
Applying IPTW	0.50 (0.43–0.58)	0.62 (0.50–0.76)	0.67 (0.58–0.79)	1.62 (1.20–2.19)	3.04 (2.01–4.60)
Restricted to the lazertinib group ^c^	0.37 (0.25–0.55)	0.54 (0.33–0.87)	0.46 (0.31–0.69)	2.12 (1.04–4.32)	2.72 (1.13–6.52)
Alternative PS model 1 ^d^	0.39 (0.27–0.55)	0.51 (0.32–0.82)	0.53 (0.38–0.75)	1.79 (0.998–3.21)	1.78 (0.84–3.76)
Alternative PS model 2 ^e^	0.46 (0.32–0.65)	0.62 (0.39–0.99)	0.63 (0.44–0.89)	1.90 (1.05–3.42)	2.09 (0.99–4.43)

PS, propensity score; HR, hazard ratio; OR, odds ratio; CI, confidence interval; PFS, progression-free survival; OS, overall survival; TTD, time to treatment discontinuation; ORR, objective response rate; DCR, disease control rate; IPTW, inverse probability of treatment weighting. ^a^ Imbalanced potential confounders were adjusted as covariates. ^b^ Propensity score was estimated using the following covariates: age, sex, smoking status, ECOG performance status, brain metastasis, previous lines of EGFR-TKI treatment, time of last EGFR-TKI treatment, and duration of previous EGFR-TKI treatment. ^c^ The lazertinib group was restricted to patients who participated in the LASER 201 and LASER 301 studies. ^d^ Propensity score was estimated using the following covariates: age, sex, smoking status, ECOG performance status, brain metastasis, type of previous EGFR-TKI treatment, time from immediate previous EGFR-TKI treatment, and duration of previous EGFR-TKI treatment. ^e^ Propensity score was estimated using the following covariates: age, sex, smoking status, ECOG performance status, brain metastasis, previous lines of EGFR-TKI treatment, type of previous EGFR-TKI treatment, time from immediate previous EGFR-TKI treatment, and duration of previous EGFR-TKI treatment.

**Table 3 cancers-16-02169-t003:** Objective response to treatment.

	Before PS Matching			After PS Matching		
	Lazertinib (*n* = 206)	External Comparator (*n* = 334)	*p*-Value	Lazertinib (*n* = 156)	External Comparator (*n* = 156)	*p*-Value
**Type of response, *n* (%)**						
Complete response	0 (0.0)	0 (0.0)		0 (0.0)	0 (0.0)	
Partial response	131 (65.5)	157 (47.0)		100 (64.1)	74 (47.4)	
Stable disease	45 (22.5)	96 (28.7)		38 (24.4)	47 (30.1)	
Progressive diseases	15 (7.5)	75 (22.5)		12 (7.7)	32 (20.5)	
Not evaluable	3 (1.5)	6 (1.8)		2 (1.3)	3 (1.9)	
Unknown, *n* (%)	6 (3.0)	0 (0.0)		4 (2.6)	0 (0.0)	
**Objective response rate**						
No. of patients, *n*	131	157		100	74	
Percentage of patients (95% CI)	65.5 (58.9–72.1)	47.0 (41.7–52.4)		64.1 (56.6–71.6)	47.4 (39.6–55.3)	
Crude OR (95% CI)	2.14 (1.49–3.07)	1.00 (Reference)	<0.001	1.98 (1.26–3.12)	1.00 (Reference)	0.003
Adjusted OR (95% CI) ^a^	1.89 (1.18–3.04)	1.00 (Reference)	0.009	1.92 (1.08–3.39)	1.00 (Reference)	0.026
**Disease control rate**						
No. of patients, *n*	176	253		138	121	
Percentage of patients (95% CI)	88.0 (83.5–92.5)	75.8 (71.2–80.4)		88.5 (83.5–93.5)	77.6 (71.0–84.1)	
Crude OR (95% CI)	2.35 (1.43–3.85)	1.00 (Reference)	<0.001	2.22 (1.20–4.12)	1.00 (Reference)	0.012
Adjusted OR (95% CI) ^a^	1.98 (1.07–3.67)	1.00 (Reference)	0.029	1.96 (0.93–4.13)	1.00 (Reference)	0.077

PS, propensity score; OR, odds ratio; CI, confidence interval. ^a^ Imbalanced potential confounders were adjusted as covariates.

## Data Availability

Access to individual participant data will not be granted, as it necessitates legitimate administrative approval. Data access is restricted in accordance with government regulations.

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
