# Peer review of "Lazertinib versus Platinum-Based Chemotherapy with Epidermal Growth Factor Receptor (EGFR)-Positive Non-Small-Cell Lung Cancer after Failing EGFR-Tyrosine Kinase Inhibitor: A Real-World External Comparator Study"

_cancers, 2024, doi:10.3390/cancers16122169_

Round 1

Reviewer 1 Report

Comments and Suggestions for Authors

I went through the manuscript entitled as “ Lazertinib Versus Platinum-Based Chemotherapy with EGFR-2 Positive Non-Small Cell Lung Cancer after Failing EGFR-TKI: 3 A Real-World External Comparator Study” by Drs Shin and Jung and co-workers.

In this real world external comparator study, authors are aiming to investigate the com-19 pared the efficacy of lazertinib and platinum-based chemotherapy in patients who had previously received EGFR-TKI treatment. Authors chooses 200 patients from the LASER 201, LASER 301, 21 LASER-PMS studies and 334 patients who received platinum-based chemotherapy after prior 22 EGFR-TKI treatment for SMC. Based on the comparisons, authors concluded that lazertinib treatment showed significantly superior efficacy compared to platinum-based chemotherapy.

In general, the authors are carefully selected the patients from the three different external studies, and fairly compare the efficacy of the treatment, the results are very clear in most of the results indicated the Lazertinib showed significant advantage over the platinum-based chemotherapy.

Albeit the weakness of the study, which authors appreciated, is that differences of the EGFR mutation between the two groups T790M vs other mutants, authors concluded that lazertinib demonstrated significantly superior efficacy compared 27 to platinum-based chemotherapy.

 There are several points below need to be addressed to clarify. 

Table S1: It is not clear what “O” all the way across the tables indicated in the table.

Line 101, and Fig1, authors indicate to select “over 18 years of age” patient for the study. However, the actual age of the patients are between 55~71 based on the Table 1, which need some adjustment.

As part of the practical limitations of the external comparison studies, one of the major weakness of the study, which authors appreciated, is that differences of the EGFR mutation between the two groups T790M vs other mutants. Are there any external data available which include T790M mutation for the study?

“This study has several limitations. Because the standard treatment for T790M-mutated NSCLC patients has been osimertinib during the period of clinical studies for lazertinib, most patients in the external group were EGFR T790M-negative or unknown, unlike the patients of the lazertinib group. In this study, the median PFS for platinum-based chemotherapy was 5.03 months (95% CI: 4.80-5.60 months), but in previous retrospective studies of second-line therapy for EGFR-mutated NSCLC and IMPRESS study in which most patients were EGFR T790M-negative, the median PFS for platinum-based chemotherapy was about 3-5.5 months [14-16].

Since this is very crucial part of the analysis of this study, the final paragraph of the discussion may not be sufficient to logically address the issue, in my view.

Table 1 “ PS matching” How authors matched the PS among the two patients population may not be clear. Are those comparison between the same PS groups comparison, or adjusted just overall PS status between the two groups?

In the table 1, It is not clear how min= max is distributed 1-10, but median can become 1, “Previous lines of systemic therapy, median (min-max)1 (1-10)” Similar concerns are the lane below “Previous lines of EGFR-TKI treatment, median (min-max)1 (1-10)”.

It seems that prior treatment between two groups are quite different between the usage of Gefinitib vs Afatinib 60.3/29.5 (Lazertinib group) 34.6/57.1 (external comparator). This also may have affected the comparison, which need to discuss in the discussion, perhaps.

Reviewer 2 Report

Comments and Suggestions for Authors

In this study, the authors compared 200 patients derived by different trials such as  LASER201, LASER301, and LASER-PMS with 334 patients treated with platinum-based chemotherapy after previous EGFR-TKI from Samsung Medical 36 Center. After propensity score matching (PSM), they enrolled 156 patients from each group. They analyzed the primary outcome progression through progression free survival al (PS),  overall survival (OS), objective response  rate (ORR), and time to treatment discontinuation (TTD) as secondary outcomes. They found that The median follow-up of PFS was 15.61 months in the lazertinib group and 21.67 months in the external control group. Specificlly, they showed that the PFS was significantly longer in patients treated with lazertinib than those treated with platinum-based chemotherapy  Lazertinib showed superior OS (32.23 43 months vs 18.73 months; adjusted HR 0.45; 95% CI, 0.29-0.69; p < 0.001), ORR (64.1% vs 47.4%), and 44 TTD (11.66 months vs 6.73 months; adjusted HR 0.54; 95% CI, 0.39-0.75; p < 0.001) to platinum-based 45 chemotherapy. In conclusion,   this retrospective, external control study showed that lazertinib had significantly better efficacy compared with platinum-based chemotherapy.

The paper is cleary written. The introduction and background arev reasonable given the promise of the paper. The tables are very comprehensive and helpful.  Minor comments  you  selected   NSCLC patients  treated platinum-based chemotherapy after previous EGFR-TKIs and in this regard, it could be useful to specife the TKIs  that are used, please add.

Round 2

Reviewer 1 Report

Comments and Suggestions for Authors

Authors have clarified and addressed my previous cencerns as satisfactory manner to clarify the issues.